# Dietary Fibre Intake and the 10-Year Incidence of Tinnitus in Older Adults

**DOI:** 10.3390/nu13114126

**Published:** 2021-11-17

**Authors:** Diana Tang, Yvonne Tran, Giriraj S. Shekhawat, George Burlutsky, Paul Mitchell, Bamini Gopinath

**Affiliations:** 1Department of Linguistics, Macquarie University Hearing, Macquarie University, Balaclava Road, Sydney, NSW 2109, Australia; yvonne.tran@mq.edu.au (Y.T.); george.burlutsky@mq.edu.au (G.B.); bamini.gopinath@mq.edu.au (B.G.); 2College of Nursing and Health Sciences, Flinders University, Adelaide, SA 5001, Australia; giriraj.shekhawat@flinders.edu.au; 3Ear Institute, University College London, London WC1X 8EE, UK; 4Tinnitus Research Initiative, 93053 Regensburg, Germany; 5Centre for Vision Research, Westmead Institute for Medical Research and Department of Ophthalmology, University of Sydney, Camperdown, NSW 2006, Australia; paul.mitchell@sydney.edu.au

**Keywords:** tinnitus, incidence, dietary fibre, older adults

## Abstract

Tinnitus is the phantom perception of sound in the ears or head that increases in prevalence as age increases. With strong evidence supporting the benefits of dietary fibre for vascular health and hearing loss, intake of dietary fibre may also have a role in the prevention of tinnitus symptoms. This longitudinal study aims to determine the association between the intake of dietary fibre and other carbohydrate nutrition variables including glycaemic index (GI), glycaemic load (GL) and total carbohydrate intakes, and incident tinnitus over 10 years. Of the 1730 participants (aged ≥50 years) from the Blue Mountains Hearing Study with complete baseline data on tinnitus symptoms and carbohydrate intakes, 536 (31%) cases of tinnitus were identified and excluded from further incidence analysis. Dietary data were collected using a validated semi-quantitative food frequency questionnaire to determine intakes of total dietary fibre and fibre contributions from cereals, vegetables, and fruit. A purpose-built database based on Australian GI values was used to calculate mean GI. Lower versus higher intakes of fruit fibre (≤3.6 g/day vs. >3.6 g/day) and cereal fibre (≤4.2 g/day vs. >4.2 g/day) were significantly associated with a 65% (HR = 1.65; 95% CI: 1.15–2.36) and 54% (HR = 1.54; 95% CI: 1.07–2.22) increased risk of developing tinnitus over 10 years, respectively. Associations between intake of other carbohydrate nutrients and incident tinnitus were mostly non-significant. In summary, our study showed modest associations between intake of dietary fibre and incident tinnitus. The protective effects of fibre, particularly insoluble fibre, could underlie observed associations by reducing the risk of tinnitus via vascular risk factors such as cardiovascular disease. Further longitudinal studies evaluating different types and sources of fibre and tinnitus risk are needed to confirm our study findings.

## 1. Introduction

Tinnitus is defined as the phantom perception of sound in the ears or head [1]. At least one in ten adults are affected by tinnitus, with the prevalence of tinnitus increasing as age increases [2,3]. Despite its high prevalence, the aetiology of tinnitus remains unclear, and various pathways including links to vascular health have been proposed [4,5]. Our prior research identified that the presence of hearing loss and dizziness symptoms are significant risk factors for tinnitus [6], while other studies have identified additional risk factors including work-related noise exposure, history of middle ear or sinus infections, severe neck injury or migraine [7,8]; history of cardiovascular disease [9]; hypertension, high cholesterol, obesity [10,11]; and smoking [3,12]. Diet may be an important modifiable risk factor in the management and prevention of tinnitus symptoms. Research evidence around nutrition and tinnitus in adults are mostly derived from a handful of cross-sectional studies from populations in the United States [13], United Kingdom [14,15], and Korea [16,17]. Some of the reported protective associations included having a healthier overall diet [13]; higher intakes of B-vitamins [15,16]; and consuming fish and other protein sources [14,15,16]. However, the lack of robust longitudinal data presents a critical gap in epidemiological research and highlights the need for long-term studies to determine the role of dietary risk factors in the development of tinnitus. 

With consideration of the potential link between tinnitus and vascular health, and strong evidence supporting a fibre-rich diet for cardiovascular diseases [18,19,20], dietary fibre may also be protective against the development of tinnitus. However, limited research of any design has investigated this association. Dietary fibre is an essential component of a healthy balanced diet, and the daily Suggested Dietary Target (SDT) to reduce the risk of chronic disease is 28 g for women and 30 g for men [21]. Dietary fibre is also widely available in the food supply, with three of the five core food groups (fruits, vegetables, and cereal/cereal products) contributing approximately 66% of total fibre intake in the diet of an average Australian aged 51–70 years [22].

Other carbohydrate variables including glycaemic index (GI), glycaemic load (GL), total carbohydrate intake, and total sugar intake are also linked to vascular health [23,24] and may similarly influence tinnitus risk. This novel study aims to address the current gaps in the research literature around diet and tinnitus risk by reporting on the associations between carbohydrate nutrient variables (dietary fibre, GI, GL, and intakes of carbohydrates) and the 10-year incidence of tinnitus in a large population-based cohort of older Australians aged ≥50 years.

## 2. Materials and Methods

### 2.1. Study Participants

The participants of the Blue Mountains Hearing Study (BMHS) have been described previously [25]. At baseline, 2015 older adults aged ≥50 years who were participating in the Blue Mountains Eye Study (BMES) II from 1997–1999 were recruited into the BMHS. Of the 2015 participants, 1730 had baseline data on tinnitus symptoms and dietary data on the carbohydrate nutrient variables. There were 536 (31%) cases of tinnitus at baseline, and these participants were excluded from further analysis. The remaining 1194 participants were followed up over 10 years (until 2007–2009). The BMHS was conducted in accordance with the Declaration of Helsinki and was approved by The University of Sydney Human Research Ethics Committee (Reference: HREC 9826).

### 2.2. Tinnitus and Hearing Assessment

The presence of tinnitus symptoms was determined by an audiologist-administered question: “Have you experienced any prolonged ringing, buzzing, or other sounds in your ears or head within the past year that is lasting for five minutes or longer?” Additional information on tinnitus risk factors were also collected, i.e., any hearing loss was determined by an audiologist in sound-treated booths using TDH-39 earphones and Australian-standardised Madsen OB822 audiometers (Madsen Electronics), and dizziness symptoms were collected via a self-administered questionnaire.

Incident tinnitus was defined as participants who reported no tinnitus symptoms at baseline (1997–1999) but reported tinnitus symptoms at either the 5- or 10-year follow-up in 2002-2004 and 2007–2009, respectively. 

### 2.3. Dietary Data

Self-reported usual dietary intakes over the last 12 months from baseline were captured using a validated semi-quantitative food frequency questionnaire (FFQ) [26]. This 145-item FFQ was modified to suit the Australian population. For each food item, a reference portion was included and a nine-category frequency scale ranging from ‘Never’ to ‘4+ per day’ was used to determine usual intake. Responses were then converted to fractions of one day and multiplied by the reference portion to determine daily intake. The FFQ included additional questions on types of breakfast cereals consumed to increase the accuracy of calculated carbohydrate nutrient variables. To calculate dietary intakes, FFQ data was linked to the Australian food composition tables 1990 (NUTTAB 90) [27]. Total fibre intake, total carbohydrate intake, and total sugar intake were calculated by multiplying the daily intake of each food item in the FFQ by the respective available amount (in grams) of these carbohydrate nutrients according to NUTTAB 90 [27] and then totalling intakes across all food items. Similar calculations were applied to determine fibre contributions from cereals, fruits, and vegetables and totalling the relevant food items. 

To calculate GI, FFQ data were coded into a customised database (DBASE IV; Borland International Inc., California, USA, 1991) that incorporated NUTTAB 90 [27] and published GI values using the glucose = 100 scale and additional GI data from the Sydney University Glycaemic Index Research Service (SUGiRS) online database [28]. Most (88.9%) GI values were sourced from published values, with the remainder (11.1%) interpolated from comparable food items. Overall dietary GI was calculated by totalling the weighted GI of each food item, where weight was a proportion of the food item’s contribution to total carbohydrate intake. Each food item’s GI was then multiplied by the amount of available carbohydrate to calculate the GL of each food item, i.e., GL = GI (%) × carbohydrate (g). Overall dietary GL was then calculated as a food item’s GL multiplied by the frequency of consumption and totalled for all food items.

### 2.4. Statistical Analysis

Analyses were performed using SAS statistical software (SAS Institute, Cary, NC, USA) version 9.4. Descriptive statistics were used to describe the participants, stratified by quartiles of fruit and cereal fibre intake. Discrete-time proportional hazard models were used to determine the associations between carbohydrate nutrients variables (expressed as quartiles of intake) and incident tinnitus over 10 years. Results are reported as hazard ratios (HR) with 95% confidence intervals (CI). HRs have been minimally adjusted for age, sex, and energy intake. Multivariate-adjusted HRs also include significant covariates for incident tinnitus in this population i.e., presence of any hearing loss (>25 dB HL) and dizziness symptoms [6], and history of middle ear infections. For variables indicating a higher incidence of tinnitus in the first quartile of intake compared to the other quartiles, a chi-squared test was performed, comparing the incidence in quartile 1 versus the combined incidence in the remaining quartiles. Binary logistic regression models were applied to nutrition variables, indicating a significant difference (*p* < 0.05) in the risk of tinnitus between quartiles of intake.

## 3. Results

Of the 1194 eligible participants at baseline, 222 (19%) new (incident) cases of tinnitus were reported over 10 years. Participant characteristics by quartiles of fruit and cereal fibre intakes are shown in Table 1. For both nutrients, the proportion of women, smoking status, and mean intakes of carbohydrates, sugar, saturated fat, and alcohol significantly differed across quartiles. For fruit fibre only, the presence of any hearing loss was also significantly different, and for cereal fibre only, the proportion of participants who were physically inactive was significantly different across quartiles.

After adjustments for age, sex, and energy intake, significant reductions in the risk of incident tinnitus were observed (Table 2). There was a 34% (HR = 0.66; 95% CI: 0.43–0.99) reduction in risk among participants in the second versus first quartile of sugar intake; a 34% (OR = 0.66; 95% CI: 0.44–0.99) reduction in risk of tinnitus among participants in the third versus first quartile of cereal fibre intake; and a 63% (HR = 0.57; 95% CI: 0.38–0.87) reduction in risk among participants in the second versus first quartile of fruit fibre intake. A stronger association between higher intakes of fruit fibre and cereal fibre, and a reduced risk of incident tinnitus were observed following additional adjustments for any hearing loss, history of middle ear infection and the presence of dizziness, i.e., participants in the second versus first quartile of fruit fibre and cereal fibre intake had a 53% (HR = 0.47; 95% CI: 0.29–0.76) and 39% (HR = 0.61; 95% CI: 0.39–0.98) reduced risk of developing tinnitus over 10 years, respectively. The additional adjustments also showed that participants in the second versus first quartile of intake of total fibre, and those in the fourth versus first quartile of intake of carbohydrates had a 40% (HR = 0.60, 95% CI: 0.37–0.96) and 45% (HR = 0.55, 95% CI: 0.34–0.90) reduced risk of incident tinnitus, respectively.

Further analyses of these carbohydrate nutrients showed that a lower intake (1st quartile) of fruit fibre and cereal fibre compared to higher intakes (quartiles 2–4 combined) significantly increased the risk of developing tinnitus by 65% (multivariable-adjusted HR = 1.65, 95% CI: 1.15–2.36) and 54% (multivariable-adjusted HR = 1.54; 95% CI: 1.07–2.22), respectively (Table 3).

## 4. Discussion

Diet may be an important modifiable risk factor for the development of tinnitus symptoms. However, most studies exploring this association have been cross-sectional. To our knowledge, this is the first longitudinal cohort study to report on the associations between carbohydrate nutrient variables and the 10-year incidence of tinnitus in an older adult population. We found that the participants with the lowest quartile of intake compared to higher intakes of fruit and cereal fibre had a significantly increased risk of developing tinnitus, by 65% and 54%, respectively. Among our participants, the lowest quartile of intake of fruit fibre was ≤4.2 g/day, and cereal fibre was ≤3.6 g/day. Compared to the general Australian population, adults aged >50 years generally consume 3.9–4.5 g fruit fibre/day and 7.6–8.3 g cereal fibre/day [22,29]. These intake levels are higher than our respective cut-offs for quartile 1 and suggest that participants within the lowest quartile of intake of fruit and cereal fibre may have poorer lifestyle habits compared to the general Australian population as well as the other study participants. This is supported by the higher proportion of current smokers observed in the lower quartiles of fruit and cereal fibre intakes. In the first quartile of intake of fruit and cereal fibre, 14.6% and 14.5% were current smokers, compared to 4.1% and 4.1% in the highest quartiles of intake, respectively. Similarly, an increasing amount of alcohol (g) is consumed by participants in the lowest quartiles of fruit and cereal fibre intakes; that is, almost double the amount of alcohol is consumed by participants in quartile 1 (16.2 g and 16.6 g, respectively) versus quartile 4 (8.3 g and 8.4 g, respectively). Participants in quartile 1 versus quartile 4 for both fruit and cereal fibre intakes also consume more saturated fat (35.0 g vs. 24.9 g; 32.6 g vs. 27.4 g, respectively) and have a larger proportion who are physically inactive (65% vs. 58%; 70% vs. 46%, respectively).

Cereal fibre may not only be protective against incident tinnitus but beneficial for overall ear health. Our prior research has shown that higher cereal fibre intake is protective against age-related hearing loss, as participants in the lowest tertile of cereal fibre intake were 70% more likely to have mild hearing loss [multivariable-adjusted OR = 1.70 (95% CI = 1.07–2.69)] [30]. Adjusting for cereal fibre intake also cancelled out significant associations between higher intakes of carbohydrates and increased risk of incident hearing loss [30]. These findings suggest that the benefits of cereal fibre for hearing loss may be related to improvements in insulin sensitivity or postprandial glycaemia [30]. Similarly, our significant finding between lower cereal fibre intake and a 42% increase in tinnitus risk may also be due to reduced insulin sensitivity. Cereals are a major source of insoluble fibre, and studies have shown that insoluble cereal fibre can significantly improve insulin sensitivity [31]. Moreover, hyperinsulinaemia due to low insulin sensitivity contributes to the development of inner ear disturbances [32,33], which could explain the observed association with 10-year incidence of tinnitus.

Although we did not distinguish between insoluble and soluble fibre in this study, insoluble fibre intake may be the underlying mechanism for observed protective associations of fibre intake against tinnitus risk. In addition to cereals, insoluble fibre is also commonly sourced from the skins of fruits and vegetables [34]. However, as common dietary practices include peeling and cooking many vegetables, the contribution of insoluble fibre from vegetables may be lower than from fruit, which are commonly consumed raw, and this could explain the largely non-significant finding with incident tinnitus.

In addition to improving insulin sensitivity, there is a wealth of evidence supporting the role of dietary fibre for vascular health [18,19,20]. For example, one systematic review and meta-analysis [18] reported that insoluble fibre and fibre contributions from cereals, fruits, and vegetables were inversely associated with cardiovascular disease risk. There is also evidence that vascular health is linked to the aetiology of tinnitus [4,5,35]. Specifically, damage to the cochlear due to microcirculation abnormalities that cause a persistent reduction in blood flow to the ear may increase the risk of developing tinnitus symptoms [35]. The established evidence supporting links between intake of dietary fibre and cardiovascular disease [18,19,20] in addition to our findings showing that low intakes of fruit and cereal fibre significantly increase tinnitus risk is supportive of a link between vascular pathways and the onset of tinnitus. Acknowledging this relationship, it is possible that our findings are the result of dietary fibre reducing the risk of vascular health indices and thereby, reducing the risk of developing tinnitus.

The burden of tinnitus extends to various aspects of an individual’s life, negatively impacting sleep, attention, and overall quality of life [36]. As the prevalence of tinnitus increases with older age [2,3], the burden from this condition will escalate. Identifying potential protective and/or therapeutic approaches to reducing the risk of developing tinnitus and managing the symptoms is warranted. Our findings suggest that dietary modifications involving increasing fruit and cereal fibre intake as part of the habitual diet are a possible strategy to reduce the burden of tinnitus. 

### Strengths and Limitations

The strengths of this study include a large representative population of older Australians to minimise the risk of selection bias and the use of a validated food questionnaire to capture usual dietary intake. We acknowledge that this study is not without limitations. Firstly, the FFQ captures self-reported usual intake in the last 12 months and, therefore, reported intakes may vary from actual intakes. Secondly, the available data within the NUTTAB 90 food composition database linked to the FFQ did not distinguish between soluble and insoluble fibre. Additional studies should consider other nutrient databases that include this information to further explore the association between types of fibre and incident tinnitus. Thirdly, we acknowledge that unaccounted/unmeasured covariates and/or other dietary variables may be influencing the reported findings. Finally, the presence of tinnitus symptoms was based on a single question and may be subject to recall bias. Thus, our findings do not suggest a causal relationship between low intakes of fruit and cereal fibre and incident tinnitus, and we strongly recommend conducting further longitudinal studies to confirm this association.

## 5. Conclusions

In summary, tinnitus is a highly prevalent condition that is expected to increase due to the ageing demographics of most countries. Preventive evidence-based strategies to reduce the burden of this condition are critical. This study is the first to report that low intakes of fruit and cereal fibre increases the risk of developing tinnitus over 10 years in older adults. These associations may reflect the benefits of dietary fibre on vascular health, whereby vascular alterations are a risk factor for tinnitus. Additional robust data such as those from large cohort studies and randomised controlled trials further exploring potential links between carbohydrate nutrition and tinnitus are needed to confirm this finding.

## Figures and Tables

**Table 1 nutrients-13-04126-t001:** Characteristics of participants stratified by quartiles of fruit and cereal fibre intake at baseline.

	Intakes of Fruit Fibre				
Characteristics ^1^	Quartile 1 (0.0–3.6 g/d)	Quartile 2 (>3.6–6.2 g/d)	Quartile 3 (>6.2–9.7 g/d)	Quartile 4 (>9.7–43.9 g/d)	*p*-Value ^2^
Women, *n* (%)	110 (49.1)	141 (63.0)	143 (63.8)	142 (63.4)	0.003
Mean age (SD), years	67.0 ± 7.1	68.2 ± 7.4	67.3 ± 7.5	67.6 ± 7.3	0.39
Any hearing loss, *n* (%)	63 (28.3)	80 (35.7)	45 (20.1)	56 (25.0)	0.002
Dizziness symptoms, *n* (%)	56 (27.1)	71 (34.0)	64 (31.4)	59 (30.0)	0.49
Current smoking, *n* (%)	32 (14.6)	18 (8.1)	13 (5.8)	9 (4.1)	0.0003
History of Type 2 diabetes, *n* (%)	25 (11.2)	24 (10.7)	24 (10.7)	18 (8.0)	0.68
History of CVD, *n* (%)	24 (10.9)	19 (8.5)	19 (8.5)	26 (11.6)	0.58
History of hypertension, *n* (%)	106 (47.3)	114 (51.1)	118 (52.7)	105 (47.1)	0.56
History of high cholesterol, *n* (%)	51 (23.9)	59 (27.6)	75 (34.6)	71 (32.6)	0.07
Obesity (BMI ≥30 kg/m^2^), *n* (%)	47 (22.6)	49 (23.6)	53 (25.4)	47 (22.4)	0.89
Physically inactive, *n* (%)	65 (31.0)	68 (33.0)	53 (24.9)	58 (28.2)	0.29
Mean Intake (SD), g/day:					
Carbohydrates Sugar Protein Saturated fat Alcohol	240 (75.4) 120 (51.3) 94.6 (29.2) 35.0 (13.3) 16.2 (20.2)	219 (71.9) 112 (45.7) 85.6 (28.3) 27.4 (11.0) 10.7 (15.2)	234 (65.1) 122 (41.7) 89.4 (28.0) 26.0 (9.90) 9.7 (15.3)	269 (81.6) 154 (52.6) 92.3 (27.4) 24.9 (10.9) 8.3 (11.6)	<0.0001 <0.0001 0.01 <0.0001 <0.0001
	**Intakes of Cereal Fibre**				
**Characteristics ^1^**	**Quartile 1** **(0.0–4.2 g/d)**	**Quartile 2** **(>4.2–6.7 g/d)**	**Quartile 3** **(>6.7–9.7 g/d)**	**Quartile 4** **(>9.7–42.2 g/d)**	***p*-Value**
Women, *n* (%)	120 (53.6)	142 (63.4)	151 (67.4)	123 (54.9)	0.006
Mean age (SD), years	67.1 ± 7.1	67.3 ± 7.1	68.2 ± 7.2	67.4 ± 7.9	0.42
Any hearing loss, *n* (%)	69 (30.8)	55 (24.7)	64 (28.6)	56 (25.0)	0.40
Dizziness symptoms, *n* (%)	56 (27.9)	61 (29.9)	64 (31.4)	69 (33.2)	0.69
Current smoking, *n* (%)	32 (14.5)	21 (9.5)	10 (4.5)	9 (4.1)	<0.0001
History of Type 2 diabetes, *n* (%)	25 (11.2)	24 (10.7)	23 (10.3)	19 (8.5)	0.80
History of CVD, *n* (%)	23 (10.3)	18 (8.1)	24 (10.7)	23 (10.4)	0.78
History of hypertension, *n* (%)	114 (51.1)	111 (49.6)	110 (49.1)	108 (48.4)	0.95
History of high cholesterol, *n* (%)	61 (28.8)	66 (30.4)	63 (29.2)	66 (30.4)	0.97
Obesity (BMI ≥30 kg/m^2^), *n* (%)	45 (21.4)	50 (24.4)	49 (23.6)	52 (24.5)	0.87
Physical inactive, *n* (%)	70 (33.0)	62 (30.4)	66 (32.0)	46 (21.6)	0.04
Mean Intake (SD), g/day:					
Carbohydrates Sugar Protein Saturated fat Alcohol	231 (81.2) 132 (55.6) 90.6 (31.9) 32.6 (14.5) 16.6 (21.1)	223 (69.1) 119 (47.8) 88.5 (25.9) 27.6 (11.1) 11.1 (14.5)	246 (73.2) 127 (48.7) 90.2 (30.0) 25.8 (10.2) 8.6 (13.1)	262 (73.9) 130 (49.0) 92.5 (25.3) 27.4 (10.6) 8.4 (13.1)	<0.0001 0.03 0.51 <0.0001 <0.0001

^1^ Number of participants for each characteristic may not add up to the total number of participants, as there are missing values. ^2^ *p*-value < 0.05 indicated statistical significance.

**Table 2 nutrients-13-04126-t002:** Associations between mean dietary GI, dietary GL, carbohydrate, sugar, fibre (expressed in quartiles), and the 10-year incidence of tinnitus.

Variable (g/Day)	*n* at Risk/Cases	Any Tinnitus	
		HR (95% CI) ^1^	HR (95% CI) ^2^
Mean dietary GL			
1st quartile (40.2–105.5)	224/57	1.0 (reference)	1.0 (reference)
2nd quartile (>105.5–129.0)	224/63	1.15 (0.77–1.71)	1.23 (0.80–1.91)
3rd quartile (>129.0–158.3)	224/45	0.79 (0.51–1.20)	0.75 (0.47–1.20)
4th quartile (>158.3–337.9)	224/57	1.08 (0.72–1.62)	0.83 (0.52–1.34)
Mean dietary GI			
1st quartile (41.1–53.5)	224/56	1.0 (reference)	1.0 (reference)
2nd quartile (>53.5–56.2)	224/56	1.07 (0.71–1.60)	1.23 (0.78–1.94)
3rd quartile (>56.2–58.7)	224/53	0.98 (0.65–1.48)	0.94 (0.59–1.51)
4th quartile (>58.7–71.8)	224/57	1.12 (0.74–1.69)	1.03 (0.65–1.64)
Mean Carbohydrates			
1st quartile (74.5–188.4)	224/63	1.0 (reference)	1.0 (reference)
2nd quartile (>188.4–231.7)	224/56	0.84 (0.56–1.27)	0.74 (0.47–1.17)
3rd quartile (231.8–280.8)	224/50	0.76 (0.51–1.15)	0.739 (0.47–1.15)
4th quartile (>280.8–577.7)	224/53	0.79 (0.53–1.18)	0.55 (0.34–0.90)
Mean Sugar			
1st quartile (17.1–91.0)	224/64	1.0 (reference)	1.0 (reference)
2nd quartile (>91.0–120.1)	224/45	0.66 (0.43–0.99)	0.64 (0.40–1.01)
3rd quartile (>120.1–154.0)	224/59	0.95 (0.64–1.41)	0.94 (0.61–1.47)
4th quartile (>154.0–350.8)	224/54	0.81 (0.54–1.20)	0.70 (0.44–1.12)
Mean Total Fibre			
1st quartile (1.82–17.8)	224/60	1.0 (reference)	1.0 (reference)
2nd quartile (>17.8–23.8)	224/47	0.70 (0.46–1.06)	0.60 (0.37–0.96)
3rd quartile (>23.8–30.6)	224/58	0.97 (0.65–1.45)	0.87 (0.56–1.37)
4th quartile (>30.6–89.3)	224/57	0.89 (0.59–1.32)	0.77 (0.49–1.21)
Mean Cereal Fibre			
1st quartile (0.0–4.2)	224/68	1.0 (reference)	1.0 (reference)
2nd quartile (>4.2–6.7)	224/48	0.67 (0.44–1.00)	0.61 (0.39–0.98)
3rd quartile (>6.7–9.7)	224/48	0.66 (0.44–0.99)	0.63 (0.40–1.01)
4th quartile (>9.7–42.2)	224/58	0.79 (0.54–1.17)	0.70 (0.45–1.09)
Mean Fruit Fibre			
1st quartile (0.0–3.6)	224/69	1.0 (reference)	1.0 (reference)
2nd quartile (>3.6–6.2)	224/44	0.57 (0.38–0.87)	0.47 (0.29–0.76)
3rd quartile (>6.2–9.7)	224/55	0.70 (0.47–1.04)	0.68 (0.43–1.06)
4th quartile (>9.7–43.9)	224/54	0.70 (0.47–1.03)	0.69 (0.44–1.08)
Mean Vegetable Fibre			
1st quartile (0.8–7.2)	224/51	1.0 (reference)	1.0 (reference)
2nd quartile (>7.2–9.7)	224/61	1.37 (0.91–2.06)	1.32 (0.82–2.11)
3rd quartile (>9.7–12.3)	224/51	0.99 (0.65–1.51)	0.97 (0.60–1.56)
4th quartile (>12.3–54.5)	224/59	1.28 (0.85–1.94)	1.19 (0.75–1.89)

^1^ Hazard ratios (HR) and 95% confidence intervals (CI), adjusted for age, sex, and energy intake. ^2^ Hazard ratios (HR) and 95% confidence intervals (CI), adjusted for age, sex, energy intake, the presence of any hearing loss (>25 dB HL), history of middle ear infection and dizziness symptoms.

**Table 3 nutrients-13-04126-t003:** Association between mean intakes of cereal and fruit fibre and the 10-year incidence of tinnitus.

	HR (95% CI) ^1^	HR (95% CI) ^2^
Mean Cereal Fibre		
Higher intakes (2nd, 3rd, 4th quartiles) >4.2 g/day	1.0 (reference)	1.0 (reference)
Lower intakes (1st quartile) ≤4.2 g/day	1.41 (1.03–1.94)	1.54 (1.07–2.22)
Mean Fruit Fibre (g/day)		
Higher intakes (2nd, 3rd, 4th quartiles) >3.6 g/day	1.0 (reference)	1.0 (reference)
Lower intakes (1st quartile) ≤3.6 g/day	1.52 (1.11–2.10)	1.65 (1.15–2.36)

^1^ Hazard ratios (HR) and 95% confidence intervals (CI), adjusted for age, sex and energy intake. ^2^ Hazard ratios (HR) and 95% confidence intervals (CI), adjusted for age, sex, energy intake, the presence of any hearing loss (>25 dB HL), history of middle ear infection and dizziness symptoms.

## Data Availability

The data presented in this study are available on request from the corresponding author. The data are not publicly available due to privacy restrictions.

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
