# Peer review of "Dietary Fibre Intake and the 10-Year Incidence of Tinnitus in Older Adults"

_nutrients, 2021, doi:10.3390/nu13114126_

Round 1

Reviewer 1 Report

The authors provide data from a prospective longitudinal cohort study, investigating the association between fiber intake and tinnitus.

The overall rationale of the study is clear, however, several confounders were ignored. Tinnitus - as a potential consequence of vascular alterations - is connected to hypertension and other components of the Metabolic Syndrome, some of which can in fact be modulated by fiber intake (obesity, T2DM). Also, high intake of fiber often represents a generally healthy lifestyle with more physical activity, higher intake of natural vitamins and lower intake of saturated fat and sugar.

Methodologically, the overall approach is fine, but obesity, hypertension and smoking need to be included as covariates for adjustment. Is there any other dietary information available? Fat intake? Protein intake? How about data on physical activity?

The quartile-wise analysis of HRs shows non-linear, rather U-shaped associations for most of the assessed dietary components, which makes the assumption of a causal relationship between fiber intake and tinnitus questionable.

I suggest a thorough re-analysis considering all available confounders for adjustment.

Reviewer 2 Report

Dear authors, an interesting paper. But due the other potential risk factor for developing tinnitus, like tobacco, alcohol consumption and overweight, it would have been interesting to follow these risks factors in your study. Dietary fibers could have benefits or counterbalance bad effects of alcohol, overweight...

Is it possible to have a description of the cohorte including sex, age, weigh, alcohol and tobacco consumption ? You mentioned currently smoking in table 1.

It would have also been interesting to better identify and distinguish the consumption between soluble and insoluble fibers.

Based on the discussion, mainly based on the effect of dietary fibers on cardiovascular health, and that is supported by the potential effect of cardiovascular problem on tinnitus appearance, in my opinion you cannot directly make a link between fibers consumption and tinnitus. In my opinion your discussion and conclusion, and of course abstract must be reconsidered with the option of a risk factor reduction. In my opinion, based on your discussion, it is only possible to assess that dietary fibers consumption may reduce a risk factor by beneficial impacts on cardiovascular health. It is important to distinguish between the reduction of tinnitus risk and the reduction of a risk factor of tinnitus. Reduction of risk factor better complies with nutritional requirements than with prevention and therapeutic treatments.
